# Predictive Model for the Assessment of Preoperative Frailty Risk in the Elderly

**DOI:** 10.3390/jcm10194612

**Published:** 2021-10-08

**Authors:** Sang-Wook Lee, Jae-Sik Nam, Ye-Jee Kim, Min-Ju Kim, Jeong-Hyun Choi, Eun-Ho Lee, Kyoung-Woon Joung, In-Cheol Choi

**Affiliations:** 1Asan Medical Center, Department of Anesthesiology and Pain Medicine, University of Ulsan College of Medicine, Seoul 05505, Korea; sangwooklee20@gmail.com (S.-W.L.); jaesik_nam@naver.com (J.-S.N.); leho@naver.com (E.-H.L.); icchoi@amc.seoul.kr (I.-C.C.); 2Asan Medical Center, Department of Clinical Epidemiology and Biostatistics, University of Ulsan College of Medicine, Seoul 05505, Korea; kimyejee@amc.seoul.kr (Y.-J.K.); minjukim@amc.seoul.kr (M.-J.K.); 3Department of Anesthesiology and Pain Medicine, College of Medicine, Kyung Hee University, Seoul 02453, Korea; choikhang@gmail.com

**Keywords:** frailty, emergency operation, elderly, hospital frailty risk score

## Abstract

Adequate preoperative evaluation of frailty can greatly assist in the efficient allocation of hospital resources and planning treatments. However, most of the previous frailty evaluation methods, which are complicated, time-consuming, and can have inter-evaluator error, are difficult to apply in urgent situations. Thus, the authors aimed to develop and validate a predictive model for pre-operative frailty risk of elderly patients by using diagnostic and operation codes, which can be obtained easily and quickly from electronic records. We extracted the development cohort of 1762 people who were hospitalized for emergency operations at a single institution between 1 January 2012 and 31 December 2016. The temporal validation cohort from 1 January 2017 to 31 December 2018 in the same center was set. External validation was conducted on 6432 patients aged 75 years or older from 2012 to 2015 who had emergency surgery in the Korean national health insurance database. We developed the Operation Frailty Risk Score (OFRS) by assessing the association of Operation Group and Hospital Frailty Risk Score with the 90-day mortality through logistic regression analysis. We validated the OFRS in both the temporal validation cohort and two external validation cohorts. In the temporal validation cohort and the external validation cohort I and II, the c-statistics for OFRS to predict 90-day mortality were 0.728, 0.626, and 0.619, respectively. OFRS from these diagnostic codes and operation codes may help evaluate the peri-operative frailty risk before emergency surgery for elderly patients where history-taking and pre-operative testing cannot be performed.

## 1. Introduction

Aging is an inevitable process that is measured by chronological age. There are no definite criteria for an age at which one becomes “elderly”, but according to the WHO, people aged 65 or older are classified as elderly. The number of elderly patients who undergo surgery has increased rapidly, and their age is increasing dramatically as the proportion of elderly in the population increases [1,2]. Some elderly patients with more serious adverse outcomes than in the usual clinical course have come to be called frail. Frailty describes decreased physiological reserves across multiple organ systems and increased vulnerability to disability, but it happens at different rates in different people; hence, there is a high risk for poor results given an apparently innocuous stimulus in geriatric patients [3]. Surgical stress can be a clinically significant issue for the frail in geriatric medicine [4]. Frailty in elderly surgical patients increases not only postoperative mortality and morbidity, but also the likelihood of experiencing postoperative complications and the tendency to incur more hospital costs [5,6,7]. In particular, pre-operative frailty in the emergency setting has a greater impact on poor clinical outcomes [8,9,10,11,12].

Pre-operative assessment of frailty can be helpful for the efficient application of hospital resources and planning treatments [13]. However, most of the previous evaluation systems for frailty, which can be complicated, time-consuming, and subject to inter-operator error, are difficult to apply in urgent situations requiring emergency surgery [14,15]. There are currently no relevant tools to measure peri-operative frailty risk for elderly patients undergoing emergency surgery. Almost all emergency operations have been performed without proper assessment of frailty risk; there was not enough time to do so, as previous methods of frailty risk assessment required many laboratory or clinical test results. In a previous study, there was a rapid frailty-risk evaluation method called Hospital Frailty Risk Score (HFRS) that used International Statistical Classification of Diseases and Related Health Problems, Tenth Revision (ICD-10) codes, which are diagnostic codes, but it was not a model designed for surgical patients [16].

Therefore, we aimed to create a predictive model to assess the pre-operative frailty risk of elderly patients by using diagnostic codes and operation codes that can be easily and quickly obtained from the electronic medical recording system in situations where pre-operative clinical information is insufficient, such as emergency surgery.

## 2. Materials and Methods

This study was a nationwide cohort study using the Korean National Health Insurance Database (KNHID) and a dataset from the electronic medical records of a tertiary academic center. This study was approved by the Ethics Committee (AMC IRB 2019-1145), and written informed consent was waived for retrospective data analysis. In this study, datasets from four cohorts were needed. Figure 1 shows the flow chart of the patients in this study.

### 2.1. Development Cohort and Temporal Validation Cohort

We extracted the development cohort of people who were hospitalized for emergency operations at a single institution between 1 January 2012 and 31 December 2016. We also extracted the temporal validation cohort of people aged 75 years and older who were admitted for emergency surgeries performed from 1 January 2017 to 31 December 2018 in the same center. In this study, emergency operations were defined as an operation that can claim emergency medical management fees for patients who were admitted to the hospital through the emergency department to undergo emergency surgery or an additional charge on the night or holiday/weekend if the surgery was performed after the evening of the week or during the holiday/weekend.

We included 1612 patients aged 75 years and older from the electronic medical records of our hospital as the development cohort, excluding those whose follow-up was lost or those who lacked a pre-operative operation code. We limited elderly patients to patients 75 years and older as the HFRS presented in the previous study was created for patients 75 years of age and older [16], and it is intended to be used in our study. The operation codes were claiming codes for claiming health insurance fees from the Korean National Health Insurance Service (KNHIS).

The authors of this study classified all operation codes extracted from the development cohort into a total of 8 Operation Groups (OG) according to surgical risk. The classification of operation codes was created by two clinical experts (SWL and EHL) using their clinical experience, pre-existing studies related to surgical risk, and the American College of Surgeons National Surgical Quality Improvement Program (ACS NSQIP) surgical risk calculators [17], and another clinical expert (JSN) independently checked and verified the classification of the operation codes. In the case of different opinions among experts in the classification of operation codes, the decision was made in the direction recognized by more experts by further reflecting the verification and opinions of other experts. Operations with a higher risk were classified as an increase from OG 1 to OG 8. For example, an operation code with low risk, such as “N7133” (Mastectomy) or “P4551” (Total thyroidectomy), was classified in Group 1, while an operation code with high risk, such as “O2033” (Resection of thoracic aorta aneurysm), was classified as Group 8. Appendix A shows the operation codes classified into 8 groups. The dataset of the development cohort included information about the operation codes, diagnostic codes, and death after surgery. The primary clinical outcome of this study was 90-day all-cause mortality, which was defined as the death rate within 90 days after surgery regardless of discharge.

### 2.2. Operation Frailty Risk Score

In the previous study, the HFRS was developed by using cluster analysis in such a way that scores were given for ICD-10 codes that were at least twice as prevalent in the frail group as in the other groups [16]. In this study, we created an Operation Frailty Risk Score (OFRS) by performing univariate and multivariable logistic regression analysis of the mortality within 90 days after surgery according to the operation codes of the 8 groups classified as described above, the HFRS score calculated from the diagnosis codes, age, and sex. A simple scoring system was developed using the penalized maximum likelihood estimates of the covariates in models that followed the method of Sullivan et al. [18]. After selecting a reference group of each variable, we used regression coefficients as weights and the distance from the reference group to generate each point value. Score 1 was defined as the effect of a 10-year increase in age. Through this analysis, we developed a risk scoring system of the predictive model for pre-operative frailty based on the mortality within 90 days after the operation. Based on this defined score, scores were assigned to each of the eight operation risk groups, age, and HFRS by comparing and analyzing the effects of each variable on mortality within 90 days after the operation. Table 1 summarizes the OFRS points for each variable.

### 2.3. External Validation on the Korean National Health Insurance Database

The KNHID cohort used in this study was extracted from the National Sample Cohort provided by KNHIS version 2.0 (NHIS-NSC v2.0). The NHIS-NSC v2.0 is a population-based cohort database containing clinical data of about one million patients, 2% of the sample data, which represents all national health insurance subscribers in South Korea. For external validation, we extracted patients aged 75 years or older from 2012 to 2015 who had emergency surgery in the NHIS-NSC v2.0. For external validation dataset I, we selected data considering the type of medical institutions and excluded data from clinics, which are the primary care providers. For external validation dataset II, we applied further restrictions considering the location of the medical institution, including the capital area of Seoul and Gyeonggi-do. We extracted information about the diagnostic code based on the ICD-10 and the operation code from NHIS-NSC v2.0 and evaluated the predicted model.

### 2.4. Statistical Analysis

Categorical variables are represented by numbers and percentages, while continuous variables are represented by means and standard deviations or median and interquartile range. We constructed univariate and multivariable logistic regression models to assess the association with other variables, including the HFRS and the 90-day mortality rate. Internal and external validations of the risk scoring system model were performed separately by measuring the calibration and discrimination ability. The c-statistic was used to estimate the predictive performance of the models. The calibration plot and Hosmer-Lemeshow goodness-of-fit statistic was used to evaluate the agreement between the observed and expected number of 90-days mortality across all strata, based on the probabilities of 90-day mortality estimated from the prediction model. We compared the prediction of OFRS for the pre-operative frailty risk in each dataset. For all analyses, a *p* < 0.05 was considered significant. All statistical analyses were completed using the “R” statistical language (R version 3.5.1, R Foundation for Statistical Computing, Vienna, Austria) and “SAS” Enterprise Guide ver. 7.1 (SAS Institute Inc., Cary, NC, USA).

## 3. Results

The data characteristics of the four cohorts are shown in Table 2. Comparing the four cohort groups reveals that the 90-day mortality rates of the four groups were different at a statistically significant level (*p*-value < 0.001). The 90-day mortality rate of the development cohort was 8.9%, whereas the 90-day mortality rate of the temporal validation cohort and the external validation cohort I and II was 8.4%, 13.8%, and 12.3%, respectively. In addition, the distribution of HFRS was also different in each data group. The HFRS distribution of cohort 1 and 2 showed a leftward skewed distribution pattern compared to the HFRS distribution of cohort 3 and 4.

We analyzed the distribution of emergency operations by surgical department received by the elderly patients, along with the analysis of the operation codes. According to the results for the distribution of emergency operations, an operation in the general surgery department was the most frequent in the development and temporal datasets, whereas the proportion of orthopedic surgeries was the highest in the external validation datasets. Appendix A shows the distribution of emergency operations by surgical department in each dataset.

In the development cohort, the distribution of HFRS scores was less than 5 points by 86.9%, and most were classified into the low-risk group. Therefore, due to this skewed distribution, risk stratification was performed by classifying the HFRS score with the new criteria instead of the risk classification suggested in the previous study. We re-categorized HFRS as low risk for 0 points, intermediate risk between 1 and 4 points, and high risk if above 5 points. The results of the logistic regression analyses of the risk scores for the 90-day mortality rate are summarized in Appendix A. As a result of the multivariable regression analysis based on the 90-day mortality rate of the HFRS, group 2, with 1 to 4 points of HFRS, had an odds ratio increased to 1.55 compared to risk group 1 with 0 points of HFRS. However, in the risk group 3, with 5 points or more of HFRS, the risk of a poor outcome increased with an odds ratio of 2.06 compared to the risk group 1. Table 1 summarizes the points assigned to each variable.

The OFRS is distributed in the range of 0 to 10, and it is skewed to the left in the distribution graph for each score in four cohorts (Figure 2). Additionally, Appendix A shows the distribution of OFRS for each surgical departments. According to the distribution of mortality rate according to OFRS, OFRS was classified as low risk if it was less than 2, high risk if it was greater than 4, and intermediate risk if it was between them (Table 3). Table 3 shows the overall predictive performance of the OFRS model proposed in this study in each cohort. In the temporal validation dataset, 237 (28.7%) were categorized as low risk, 355 (43.0%) as intermediate risk, and 234 (28.3%) as high risk. In the external validation dataset I, 644 (13.8%) were categorized as low risk, 3027 (64.9%) as intermediate risk, and 993 (21.3%) as high risk. In external validation dataset II, 251 (14.2%) were categorized as low risk, 1132 (64.0%) as intermediate risk, and 385 (21.8%) as high risk. From the results, it can be seen that the OFRS of the intermediated risk is more distributed in the external validation dataset than in the development and internal validation dataset. Different distributions of OFRS in these datasets affect predictive performance, and calibration performance declines.

### Internal and External Validation

We calculated the OFRS by using the operation code and HFRS calculated from the diagnosis code in each cohort. Table 3 shows the prediction performance of OFRS in each cohort. The c-statistic for internal validation of the OFRS to predict 90-day mortality was 0.682, while the c-statistic for OFRS in the temporal validation cohort and the external validation cohort I and II was 0.728, 0.626, and 0.619, respectively. Figure 3 shows the calibration of the developed OFRS model to predict outcomes in each cohort. These graphs are calibration plots showing the relationship between the real values and the predicted values of the developed OFRS model for 90-day mortality (Figure 3). Figure 4 shows the relationship between these OFRS and the 90-day mortality rate by plotting the 90-day mortality rate according to the risk scores in each validation cohort. It can be seen that the mortality rate increases as the risk scores increase.

## 4. Discussion

The main findings of this study are that OFRS, developed by using diagnostic codes and operation code information, can help predict the 90-day postoperative mortality rate, one of the indicators for pre-operative frailty when elderly patients undergo emergency surgery. Although functional improvement and decline have recently been suggested as important outcomes in elderly patients undergoing surgery, 90-day all-cause mortality rate, which has been dealt with a major postoperative clinical outcome, was set as the primary outcome in our study [19]. In the national validation cohort used for external validation, those with a higher OFRS score had higher rates of 90-day mortality, although the discriminative ability of the predictive model was low.

For elderly patients undergoing emergency surgery, pre-operative risk assessment is very important in clinical practice. In elderly patients, a high prevalence of frailty is likely to lead to postoperative adverse outcomes and vulnerability to surgical stress [4,13]. Two main models previously suggested for evaluating frailty are the phenotype model [20], and the cumulative deficit model [21]. The phenotype model proposed by Fried and colleagues was developed with five phenotypes that are largely related to frailty, and they defined “frail” as when the phenotype has more than three factors. The cumulative deficit model was defined as the proportion of each variable to the total deficits related to frailty, and they showed the correlation between the index and adverse outcomes by evaluating frailty based on this model [22]. All of these models are too complex and difficult to apply in an actual clinical setting, as it is not easy to obtain the values of each variable.

In patients undergoing emergency surgery, there have been previous studies that have tried to measure the pre-operative frailty and to determine its correlation with the postoperative outcome [8,9,10,11]. Pre-operative frailty in emergency surgical settings mostly increased the risk of postoperative mortality and longer hospital stays. Some of the previous studies on frailty in emergency surgical settings have measured frailty using the Clinical Frailty Scale [9,12,15]. It is a relatively easy and fast frailty measurement tool in the clinical setting, but it has limitations in terms of inter-operator reliability [23]. Some other studies measured pre-operative frailty in emergency surgical patients by using assessment tools such as the Modified Fried’s Frailty Criteria, the Modified Frailty Index-11, the FRAIL scale, the Triage Risk Screening Tool, and the Share-Frailty Index, and tried to determine the relationship between the measured frailty and the clinical outcome [24,25]. However, all of these tools are difficult to apply when the patient is unconscious or without present caregivers, as the methods previously proposed rely on questionnaires or require direct tests such as measurement of grip strength or walking speed.

There was a previous attempt to measure the frailty risk by using hospital electronic medical records, called HFRS, and it may be useful for measuring the frailty risk for application in acute care settings [16]. However, as the HFRS is a model developed for patients hospitalized through an emergency department, there are some limitations to applying it to surgical patients. Therefore, in order to overcome these limitations, more generally applicable frailty measurement tools are required in the emergency surgical setting. A recent study showed that both pre-operative frailty and operative stress increased postoperative mortality [6,26]. Thus, the factors of operative stress must be reflected in the pre-operative frailty risk assessment of surgical patients.

Therefore, we created an integrated frailty risk assessment tool that reflects these surgical stress factors. The OFRS proposed in this study has the advantage that it can be applied easily and quickly in emergency situations as it utilizes diagnostic codes and operation code information that can be automatically extracted from the hospital database. It is an automatic risk score that can be systematically obtained by using information from both codes without relying on the subjectivity of the clinician. It is a very useful model for elderly patients that need emergency surgery as it does not require clinical information other than the diagnostic and operation codes. Additionally, we made it possible to use the risk predictive model more reliably with wide applicability by using the insurance claiming code, which is one of the criteria commonly used by all hospitals, rather than inspection items with different standards for different hospitals.

The c-statistics of the final model in the temporal validation cohort and both external validation cohorts are 0.728, 0.626, and 0.619, respectively, in predicting the clinical outcome. Thus, our model did not show a better prediction performance than the previous model for acute-care settings showing c-statistics ranging from 0.54 to 0.73 [14,27]. Similar to the previous study [16], our scores did not have strong discriminative ability as there are unpredictable factors that affect individual outcomes in emergency situations. However, it is a predictor, which is designed for risk stratification, worth considering for application in an emergency situation where it is difficult to predict a patient’s clinical outcomes.

As the clinical setting changes according to the size and location of the hospital, it affects the clinical outcome [28,29]. In this study, the external validation of the OFRS was corrected according to the clinical environment change according to the size of the hospital and the location of the hospital. External validation dataset I extracted only hospital-level data, excluding data from clinics, and external validation dataset II applied further restrictions considering the location of the medical institution by extracting data from the capital area, geographically close to the hospital from which the dataset used to develop OFRS was extracted. In the two external datasets, the c-statistics of this model did not show significant gains in prediction performance, despite corrections according to hospital size and region. However, even within the same region, there is a considerable difference in the size of hospitals and the severity of patients, so it is expected that better predictive performance can be achieved if corrections are made for these areas in future studies.

The clinical significance of this study is that if the information about the patient is extremely limited in an urgent situation, such as emergency surgery, the patient’s risk can be grasped in advance by using the hospital’s computerized system quickly and promptly. The strength of our model is that no test results are required before surgery. There is also no need for complicated calculations to estimate the risk. As far as we know, this study is the first to predict the pre-operative risk by using only diagnostic and operation codes for surgical patients.

This study has limitations in several areas. First, it cannot safely be generalized to other clinical settings and other locations, as this model was constructed from the clinical data of a single center. As there may be differences in postoperative outcomes depending on the hospital environment, it is necessary to verify the model in various medical institutions and upgrade the model in order to have more value. In addition, developing a predictive model based on data from as many different medical institutions as possible in future studies could be a useful pre-operative risk-predictive model with broader applicability. Furthermore, it was not known whether this predictive model could be applied to other countries, especially other races, as it was developed in South Korea, which is known as an ethnically homogenous country. As the operation code in this study is the insurance claiming code commonly used in South Korea, it is difficult to apply the operation code to other countries. In order to apply this model to other countries, operation codes should be the codes that are commonly used worldwide, such as the ICD-Procedure Coding System code [30]. Therefore, future studies using this worldwide operation code for wider usability will need to verify whether the model is validated globally by using data from different races and countries. Second, another limitation of this study is that, although three clinical experts classified the OG by referring to their clinical experience, pre-existing risk prediction models, and previous studies on clinical outcomes, the subjective opinion of the clinician influenced the classification of the OG. It is expected that better results will be produced if more clinical experts participate in classifying the OG in future studies or if it is performed in a more objective fashion [31]. Additionally, in this study, the risk of all emergency surgery was classified as one criterion. However, it is difficult to classify the risk simply based on one criterion, as there may be a difference in risk according to the type of emergency operations. Therefore, in future studies, subdivision of the risk classification according to the type of emergency operation might increase predictive power. Third, another limitation of our study was that it was retrospective. Therefore, further works are needed to validate the performance and clinical usability of our predictive model in prospective multi-center studies.

## 5. Conclusions

In conclusion, the OFRS using ICD-10 diagnostic and operation codes may help to evaluate the peri-operative risk of elderly patients in emergency surgery where history-taking and pre-operative testing cannot be performed. It is expected that additional studies will broaden the applicability of the score, and the use of this score can help with decision making in various clinical settings.

## Figures and Tables

**Figure 1 jcm-10-04612-f001:**
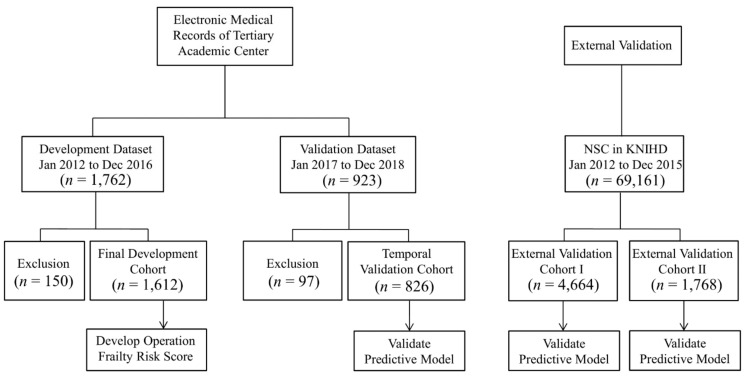
Diagram of the study dataset analysis. Four cohort datasets for the development and validation of predictive models. NSC, Normal Sample Cohort; KNHID, Korean National Health Insurance Database.

**Figure 2 jcm-10-04612-f002:**
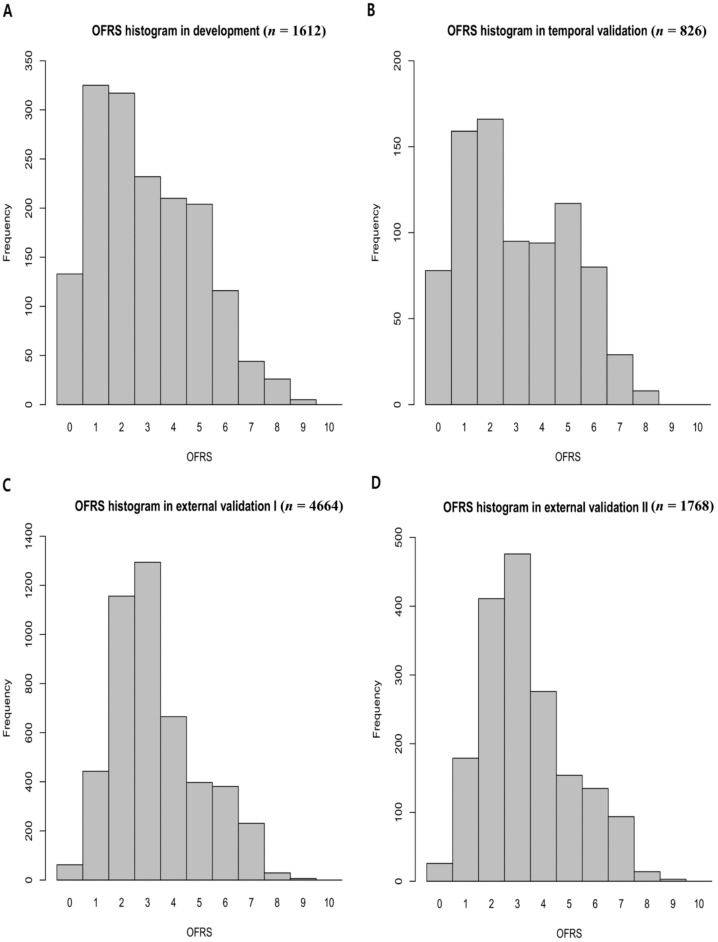
Distribution of the OFRS in (**A**) development cohort, (**B**) temporal validation cohort, (**C**) external validation cohort Ⅰ, and (**D**) external validation cohort Ⅱ. The OFRS is distributed in the range of 0 to 10. OFRS, operation frailty risk score.

**Figure 3 jcm-10-04612-f003:**
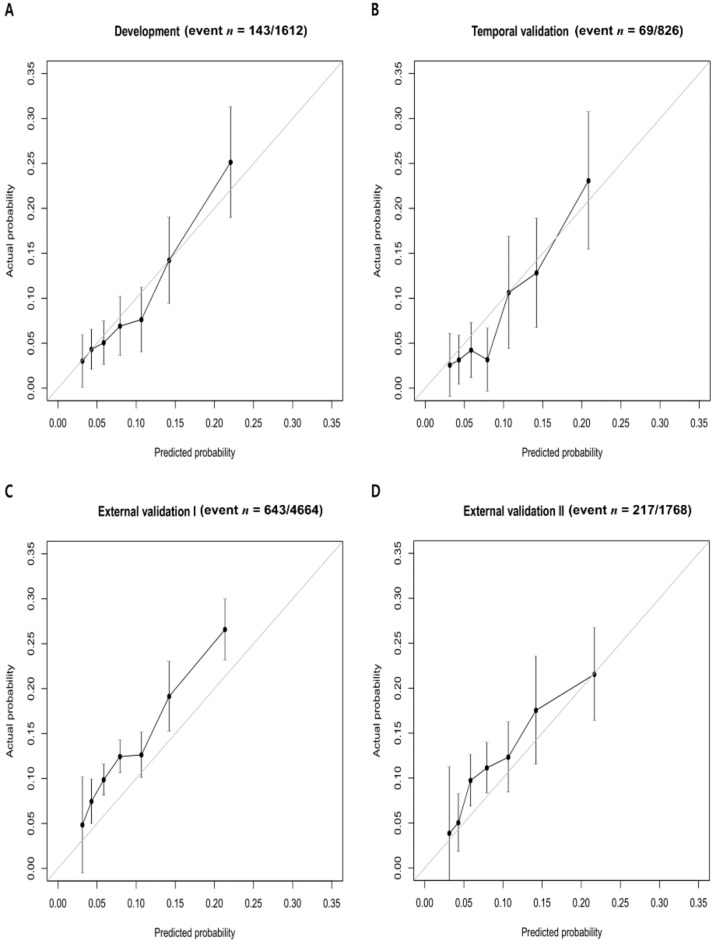
Calibration plot of OFRS in (**A**) development cohort, (**B**) temporal validation cohort, (**C**) external validation cohort Ⅰ, and (**D**) external validation cohort Ⅱ. OFRS, operation frailty risk score.

**Figure 4 jcm-10-04612-f004:**
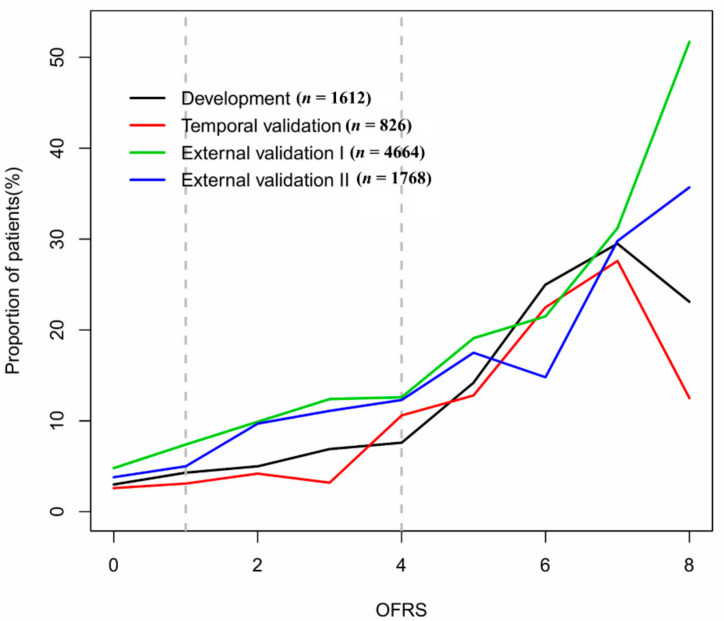
Association between OFRS and 90-day mortality by plotting the 90-day mortality rate according to the risk scores in each cohort. OFRS, operation frailty risk score.

**Table 1 jcm-10-04612-t001:** Operation frailty risk scoring system for prediction of 90-day mortality.

Variables	Categories	Point
Age	75–79	0
	80–89	1
	≥90	2
HFRS	0	0
	1–4	1
	≥5	2
Operation Group	Group 1	0
	Group 2	1
	Group 3	0
	Group 4	2
	Group 5	4
	Group 6	4
	Group 7	4
	Group 8	6

HFRS—hospital frailty risk score.

**Table 2 jcm-10-04612-t002:** Characteristics of the four cohorts.

	Cohort 1	Cohort 2	Cohort 3	Cohort 4	*p* Value
N	1612	826	4664	1768	
Age, years	78 (76–82)	78 (76–82)	80 (77–84)	80 (77–84)	<0.001
Male	862 (53.5)	445 (53.9)	2777 (59.5)	1047 (59.2)	<0.001
90-day death	143 (8.9)	69 (8.4)	643 (13.8)	217 (12.3)	<0.001
HFRS					<0.001
0	718 (44.5)	416 (50.4)	413 (8.9)	186 (10.5)	
1–4	683 (42.4)	335 (40.6)	2033 (43.6)	780 (44.1)	
≥5	211 (13.1)	75 (9.1)	2218 (47.6)	802 (45.4)	

Data are presented as the median (interquartile range) or number (percentage). Cohort 1, development cohort; Cohort 2, temporal validation cohort; Cohort 3, external validation cohort Ⅰ; Cohort 4, external validation cohort Ⅱ; HFRS—hospital frailty risk score.

**Table 3 jcm-10-04612-t003:** Characteristics and prediction performance of OFRS for 90-day mortality in each cohort.

		Cohort 1	Cohort 2	Cohort 3	Cohort 4
OFRS	Low risk (0–1)	458 (28.4)	237 (28.7)	644 (13.8)	251 (14.2)
	Intermediate risk (2–4)	759 (47.1)	355 (43.0)	3027 (64.9)	1132 (64.0)
	High risk (≥ 5)	395 (24.5)	234 (28.3)	993 (21.3)	385 (21.8)
Discrimination ability, c-statics (CI)		0.682 (0.635–0.728)	0.728 (0.665–0.791)	0.626 (0.602–0.649)	0.619 (0.580–0.658)
Calibration ability, Hosmer and Lemeshow Test	χ^2^	3.80	4.88	102.63	20.47
	DF	5	5	5	5
	*p* value	0.579	0.430	<0.001	0.001

Data are presented as the number (percentage). OFRS—operation frailty risk score; Cohort 1, development cohort; Cohort 2, temporal validation cohort; Cohort 3, external validation cohort Ⅰ; Cohort 4, external validation cohort Ⅱ; CI—confidence interval; and DF—degrees of freedom.

## Data Availability

The data presented in this study are available on request from the corresponding author (Kyoung-Woon Joung).

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
