# Peer review of "Predictive Model for the Assessment of Preoperative Frailty Risk in the Elderly"

_jcm, 2021, doi:10.3390/jcm10194612_

Round 1

Reviewer 1 Report

Eear authors

This is a very intersting study. Pre-operative risk models are highly relevant in older patients undergoing surgery.

The results of this study are presented well, and the discussion is also interesting to read from a clinical point of view. In addition to the points addressed in the discussion, I recommend also to mention the fact that another very important outcome in older patients undergoing surgery (besides mortality and length of hospital stay) is functional improvement or decline. One recent study highlighting this issue: J Am Geriatr Soc 2021 Feb;69(2):500-505. doi: 10.1111/jgs.16927. 

Author Response

  • Thank you for the good suggestion. What you mentioned was added to the discussion part of the manuscript. You can see that in line 4 of page 13. The above paper was cited as the reference in our study.

Reviewer 2 Report

The title should be shortened for conciseness.

Introduction

The whole introduction follows a logical pattern, and concludes in your aim/hypothesis - great.

Methods:

"Uncertainties and disagreements were resolved by consensus in discussion with experts" - please clarify.

Results

Fig. 2: Please add a x-axis and make the font bigger.

All figures need an explanatory figure legend. The figure should be designed in a way that the legend alone can make the reader understand each figure.

Tab. 1: Are there significant differences for each of your 8 parameters?

Tab. 3 should be mentioned right in the beginning or in the methods.

Please explain the importance of Tab. 4 in detail.

Also, the overall number of main figures and tables should be reduced. Some tables might go into supplementary.

The OFRS should be grouped according to the type of emergency operation performed. Can you provide that?

Discussion

Can you please add ideas regarding different kinds of emergency operations and the different applicability of your approach.

Author Response

Response to Reviewer 2 Comments

The title should be shortened for conciseness.

  • The title was revised as follows. If you give me a good opinion, I will reflect it. Thank you.

“Predictive model for preoperative frailty risk using diagnostic codes and operation codes in elderly patients undergoing an emergency operation: a national cohort study”

Introduction

The whole introduction follows a logical pattern, and concludes in your aim/hypothesis - great.

  • Thank you very much for your good compliment.

Methods:

"Uncertainties and disagreements were resolved by consensus in discussion with experts" - please clarify.

  • Thank you for your opinion. I revised the above sentence as follows. You can see that in line 8 of page 6.

“In the case of different opinions among experts in the classification of operation codes, the decision was made in the direction recognized by more experts by further reflecting the verification and opinions of other experts.”

Results

Fig. 2: Please add a x-axis and make the font bigger.

  • As you mentioned, Figure 2 has been modified. In the revised paper, Figure 2 was changed to Figure S1. Thanks for your comment.

All figures need an explanatory figure legend. The figure should be designed in a way that the legend alone can make the reader understand each figure.

  • I added an explanatory figure legend to each picture as below. You can see the changed figure legend on page 28. If you have any additional opinions, I would appreciate it.

Figure 1: Diagram of the study dataset analysis. This shows the composition of each cohort dataset for the development and validation of predictive models in our study.

Figure 3(Figure 2): Distribution of the OFRS in (A) development cohort, (B) temporal validation cohort, (C) external validation cohort â… , and (D) external validation cohort â…¡. The OFRS is distributed in the range of 0 to 10, and it shows the distribution graph for OFRS in four different cohorts.

Figure 4(Figure 3): Calibration plot of OFRS in (A) development cohort, (B) temporal validation cohort, (C) external validation cohort â… , and (D) external validation cohort â…¡. These graphs are calibration plots showing the relationship between the real values and the predicted values of the developed OFRS model for 90-day mortality.

Figure 5(Figure 4): Association between OFRS and 90-day mortality in each cohort. This shows the relationship between these OFRS and the 90-day mortality rate by plotting the 90-day mortality rate according to the risk scores in each validation cohort.

Tab. 1: Are there significant differences for each of your 8 parameters?

  • As mentioned in the result section of the text, there was a difference in the 90-day mortality rate among each cohort. In addition, the distribution of HFRS was also different in each data group. The contents described above were added to the manuscript in the beginning part of the result. You can see that in line 7 of page 10.

Tab. 3 should be mentioned right in the beginning or in the methods.

  • Thank you for your opinion. Table 3 (Table 2 in revised manuscript) is additionally mentioned in the OFRS part of the method. You can see that in line 13 of page 7.

Please explain the importance of Tab. 4 in detail.

  • Table 4 was changed to Table 3 in revised manuscript. The following contents were added to the result part of the manuscript. You can see that in line 9 of page 11.

“Table 3 is a table showing the overall predictive performance of the OFRS model proposed in this study in each cohort. … From the results, it can be seen that the OFRS of the intermediated risk is more distributed in the external validation dataset than in the development and internal validation dataset. Different distributions of OFRS in these datasets are affecting predictive performance and calibration performance decline.”

Also, the overall number of main figures and tables should be reduced. Some tables might go into supplementary.

  • Figure 2 and Table 2 were moved to the supplementary. Figure 2 was change to Figure S1 and Table 2 was changed to Table S1. Thank you for your opinion.

The OFRS should be grouped according to the type of emergency operation performed. Can you provide that?

  • If the type of emergency surgery means each surgical department, the distribution of OFRS scores for each surgical department was drawn as a graph and attached to the supporting material (Figure S2). It also mentioned that in the manuscript. You can see that in line 6 of page 11. Each OFRS for individual emergency operation has already been presented as a table in the supplement. Thank you for the good suggestion.

Discussion

Can you please add ideas regarding different kinds of emergency operations and the different applicability of your approach.

  • Depending on the type of emergency surgery, various postoperative outcomes are shown. In this study, the risk of all emergency surgery was classified as one criterion, but there may be a limitation in that it is difficult to classify the risk simply based on one criterion because there may be a difference in risk according to the type of surgery. Therefore, in future studies, I think it is a way to increase predictive power if the risk classification according to the type of surgery is subdivided. As you mentioned, and the contents described above were added to the discussion part of manuscript as limitations of this study. You can see that in line 10 of page 17. Thank you for your good opinion.

Round 2

Reviewer 1 Report

Thank you for the revised manuscript, I see that my comment is now addressed in the discussion.

I have one more comment/suggestion for line 318: "Additionally, in this study, the risk of all emergency surgery was classified as one criterion. However, it is difficult to classify the risk simply based on one criterion because there may be a difference in risk according to the type of emergency operations".

Line 321: Please avoid writing in the first person. Suggestion:  "Therefore, in future studies, subdivision of the risk classification according to the type of emergency operation might increase predictive power".

Author Response

Point1: Thank you for the revised manuscript, I see that my comment is now addressed in the discussion.

I have one more comment/suggestion for line 318: "Additionally, in this study, the risk of all emergency surgery was classified as one criterion. However, it is difficult to classify the risk simply based on one criterion because there may be a difference in risk according to the type of emergency operations".

Line 321: Please avoid writing in the first person. Suggestion:  "Therefore, in future studies, subdivision of the risk classification according to the type of emergency operation might increase predictive power".

Response 1: Thank you very much for your good suggestion. I revised the manuscript as you suggested. You can check the revised part on the 10th line of page 16 (line 316).

Reviewer 2 Report

The title is still way too long. Recommendation: Predictive model for the assessment of preoperative frailty risk in the elderly

or something similar.

Figure legends should not be too obviously guiding the author. "This shows..." is not necessary, for instance.

Table 1 should directly contain statistical differences, if there are any. Make it easier for the reader.

Author Response

Point1: The title is still way too long. Recommendation: Predictive model for the assessment of preoperative frailty risk in the elderly

or something similar.

Response1: Thank you for the good suggestion. I revised the title of the paper as you suggested.

Point2: Figure legends should not be too obviously guiding the author. "This shows..." is not necessary, for instance.

Response2: Thank you for your good opinion. I revised the figure legends as you suggested. Except for too detailed explanations in the figure legends, some of them were added to the text to concisely express the figures. You can check the revised manuscript on page 11 (line 211 to 213) in result section and figure legends on page 27

Point3: Table 1 should directly contain statistical differences, if there are any. Make it easier for the reader.

Response3: Thank you for your good opinion. Each p-value was added to Table 1 to reveal the significance of statistical differences. Additionally, in the case of age and gender, there was no difference between cohort 1 and 2, but there was a statistically significant difference from cohort 3 and 4. The p-value value represents the statistical significance level as to whether the four cohorts are the same, which was found to be less than 0.001. Therefore, the content that there was no significant difference between age and gender mentioned in the previous manuscript could be confusing, so it was deleted. You can check the revised text on the 3rd line of page 9 (line 166) in result section and the revised Table 1 on page 29.
